# Effects of Capsinoids on Daily Physical Activity, Body Composition and Cold Hypersensitivity in Middle-Aged and Older Adults: A Randomized Study

**DOI:** 10.3390/nu12010212

**Published:** 2020-01-14

**Authors:** Keiichi Yokoyama, Yosuke Yamada, Yasunori Akamatsu, Yasuko Yoshinaka, Akiko Yamamoto, Tomonori Koizumi, Kana Ohyama, Katsuya Suzuki, Masaki Hashimoto, Hitoshi Sato, Misaka Kimura

**Affiliations:** 1Institute for Active Health, Institutes of Interdisciplinary Research, Kyoto University of Advanced Science, 1-1 Nanjo Otani, Sogabe-cho, Kameoka-city, Kyoto 621-8555, Japan; yamada.yousuke@kuas.ac.jp (Y.Y.); kimura.misaka@kuas.ac.jp (M.K.); 2Nonprofit Organization Genki-up AGE Project, Kameoka-city, Kyoto 621-8555, Japan; yoshinaka.yasuko@kuas.ac.jp; 3Center for Faculty Development, Kyoto University of Advanced Science, 1-1 Nanjo Otani, Sogabe-cho, Kameoka-city, Kyoto 621-8555, Japan; akamatsu.yasunori@gmail.com; 4Department of Endocrinology, Metabolism, and Hypertension Research, Clinical Research Institute, National Hospital Organization Kyoto Medical Center, 1-1 Mukaihata-cho, Fukakusa, Fushimi-ku, Kyoto City, Kyoto 612-8555, Japan; 5Ajinomoto Co., Inc., Institute of Food Sciences & Technologies, 1-1 Suzuki-cho, Kawasaki-ku, Kawasaki City, Kanagawa 210-8681, Japan; akiko_yamamoto01@ajinomoto.com (A.Y.); tomonori_koizumi@ajinomoto.com (T.K.); katsuya_suzuki@ajinomoto.com (K.S.); 6Ajinomoto Co., Inc., Task Force for Nutrition Strategy, 15-1, Kyobashi 1-chome, Chuo-ku, Tokyo 104-8315, Japan; kana_oyama@ajinomoto.com; 7Ajinomoto Co., Inc., Direct Marketing Department, 15-1, Kyobashi 1-chome, Chuo-ku, Tokyo 104-8315, Japan; masaki_hashimoto@ajinomoto.com; 8Ajinomoto Co., Inc., Quality Assurance Department, 15-1, Kyobashi 1-chome, Chuo-ku, Tokyo 104-8315, Japan; hitoshi_sato@ajinomoto.com

**Keywords:** capsinoids, brown adipose tissue, physical activity, percent body fat, visceral fat, energy metabolism, energy expenditure, chilly sensation, brain inflammation, clinical

## Abstract

Sedentary/inactive lifestyle leads middle-aged and older adults to metabolic syndrome and frailty. Capsinoids from nonpungent chili pepper cultivar have been reported to reduce body fat mass, promote metabolism, and improve unidentified complaints of chills. Additionally, they have an anti-inflammation effect; therefore, we hypothesized that continuous oral ingestion of capsinoids alleviates age-related inflammation in the brain and improves the physical activity (PA) in middle-aged and older adults. In our double-blind human study, 69 participants (17 male, 52 female; mean age: 74.1 ± 7.7 years; range: 52–87 years) were administered either 9 mg of capsinoids which were extracted from pepper fruit variety CH-19 Sweet (*Capsicum anuum* L.) (CP group), or a placebo (PL group) daily over a 3 month period. In an animal study, PA and inflammation-related mRNA expression in the brain were examined in 5-week (young) and 53-week (old) aged mice fed a diet with or without 0.3% dihydrocapsiate, a type of capsinoids, for 12 weeks. In a human study, capsinoids intake did not increase the amount of light-to-moderate PA less than 6.0 metabolic equivalents (METs) (CP: 103.0 ± 28.2 at baseline to 108.2 ± 28.3 at 12 weeks; PL: 104.6 ± 19.8 at baseline to 115.2 ± 23.6 at 12 weeks, METs × hour/week); however, in participants exhibiting an inactive lifestyle, it showed significant increase (CP: 84.5 ± 17.2 at baseline to 99.2 ± 24.9 at 12 weeks; PL: 99.7 ± 23.3 at baseline to 103.8 ± 21.9 at 12 weeks). The energy expenditure in physical activity also improved in the inactive CP group (CP: 481.2 ± 96.3 at baseline to 562.5 ± 145.5 at 12 weeks; PL: 536.8 ± 112.2 at baseline to 598.6 ± 127.6 at 12 weeks; kcal/day). In all participants, CP showed reduced waist circumference, percent body fat, and visceral fat volume; in addition, chills were eased in subjects aged 80 years and older. The older mice fed capsinoids showed increased locomotion activity, decreased inflammation, and oxidative stress in the brain. The results suggest that the continuous oral ingestion of capsinoids gains PA through anti-inflammation effect in the brain as well as reduces fat accumulation and chills in inactive and older humans.

## 1. Introduction

Frailty impairs a healthy lifespan, and the core element of frailty is a decline in physical function, such as muscular strength and walking ability [1]. Furthermore, frailty has come to be recognized as a broad health problem that involves not only physical but also psychological aspects, such as diminished cognitive function and depression, and social problems, such as withdrawal and isolating behaviors [2]. Although frailty is a problem mainly in older age, studies have also found relationships between hypertension and cerebrovascular dementia [3], and between diabetes and Alzheimer’s disease [4]. Lifestyle-related diseases in middle age can also increase the risk of frailty; hence, starting prevention of frailty is critical in middle age as well as for older adults.

It is widely known that physical activity (PA) positively contributes to health, and an inactive lifestyle can lead to health problems [5,6,7]. PA controls obesity and other lifestyle-related diseases during middle age, and also lessens the risk of muscle weakness, decreased balancing capacity, falls, and fractures in the elderly [8,9]. A previous study indicates that physical activity level and activity energy expenditure are negatively associated with age after 52 years old, although they are not associated with age between 18 and 52 years old [10]. Numerous reports have examined the relationship between health and time spent sedentary (time spent sitting, watching television, driving, performing office work, etc.), and many of these reports have identified correlations between sedentary time and the onset of pathologies including diabetes and cardiovascular disease, as well as mortality risk [11]. PA can help in improving not only physical problems but also psychological problems such as Alzheimer’s disease and depression, as well as overall improved quality of life (QOL) [12,13]. As such, increasing one’s amount of PA is believed to be essential for maintaining good health, particularly from middle age onward.

Capsiate, dihydrocapsiate, and nordihydrocapsiate (referred to as capsinoids) are analog compounds of capsacin, a physiologically active heat compound, and were discovered from non-pungent chili pepper cultivar, CH-19 sweet, by Yazawa and his colleagues [14,15]. Physiological actions of capsinoids reported to date include an energy-metabolism-promoting effect, a body-heat-production-promoting effect and a body-fat-mass-reducing effect that has been confirmed in mice and humans [16,17,18,19]. For this reason, the ingestion of capsinoids is believed to reduce body fat mass in overweight individuals and reduce the risk of onset of lifestyle-related diseases.

The aging process has been shown to be associated with an increase in the proinflammatory status of organisms, and it is proposed that age-associated increase in chronic inflammation is a highly significant risk factor for age-associated disorders (called the inflamm-aging theory) [20]. The brain is not an exception; chronic inflammation is found in the human aging brain, and it is considered to be responsible for age-associated neurodegenerative disease [21]. Artificially induced brain inflammation was also found to suppress spontaneous activity in rats and mice; therefore, age-related chronic inflammation in the brain is supposed to lead to decreased physical activity in middle-aged and older adults [22,23]. Nevertheless, inflammation-mitigating agent recovered the locomotor activity impaired by artificially induced inflammation [22]. There are some findings that capsinoids have anti-inflammation pharmacological effect in vivo; mice treated with nordihydrocapsiate and dihydrocapsiate showed less peripheral inflammation [24,25]. Although there are no reports which elucidate whether capsinoids reduce inflammation in the central nervous system, considering these, we presumed that capsinoids alleviate age-related brain inflammation and increase PA in middle-aged and older adults. Cumulating research shows that brain inflammation is also a possible factor for geriatric depression [26,27]. Assuming that capsinoids have an anti-inflammation effect in the human brain, it may be expressed as a positive change of mood states.

In addition, women who often complain of feeling cold (chills or cold hypersensitivity) have been reported to suffer from swelling in the lower extremities, dizziness when standing up, stiff shoulders, and fatigue [28]. However, some improvement was observed following ingestion of capsinoids with respect to subjective complaints concerning chills by young women, and they were also shown to improve accompanying indefinite complaints such as dull head pain and shoulder stiffness [29].

We therefore hypothesized that (1) continuous oral intake of capsinoids for three months would increase PA in middle-aged and older adults, and the effect would be more pronounced in participants with sedentary lifestyles, (2) capsinoids would reduce visceral fat accumulation that would cause lifestyle-related diseases, especially in overweight participants, (3) capsinoids would alleviate the age-related brain inflammation and lead to improved PA, and (4) capsinoids would also attenuate cold hypersensitivity. For these aims, we conducted human and mice research.

## 2. Materials and Methods

### 2.1. Human Study

#### 2.1.1. Trial Design

This study was conducted as a randomized parallel-group comparison test using a double-blind dosing method, two-arm with 1:1 allocation ratio. Subjects were assigned to either a group receiving an experimental food containing capsinoids (capsinoids group, CP) or a group receiving a placebo food (placebo group, PL).

#### 2.1.2. Participants and Ethical Considerations

The subjects of this study were people aged 50 years or older who were living in two cities of Kyoto Prefecture, Japan. There were 94 applications for participation in this study, from which 80 people were selected, excluding several individuals who canceled or who fell under one or more of the study exclusion criteria described below. We decided beforehand the sample size of each group as 40 from the calculation in g*power. In this calculation, the rate of change in our intervention in cognition and mental health in older adults (unpublished data) and 5% of α error prob, 70% of power, were used. Of our subjects, 75 completed the study protocol, among whom 69 (17 males and 52 females; mean age: 74.1 ± 7.7 years, range: 52–87 years) were targeted for analysis after excluding those who fell under one or more of the exclusion criteria regarding frequency of ingesting the experimental food. Subjects for this study were selected based on the following exclusion criteria: (1) persons who were allergic to chili peppers, (2) persons who regularly consumed food products reported to have an effect on energy metabolism (catechins, sesamin, capsaicin, etc), (3) persons who regularly consumed food products reported to have effects on maintaining body temperature (monoglucohesperidin, gingerol, etc), (4) persons whose body mass index (BMI; body weight (kg)/body height (m)/body height (m)) was less than 18.5 kg/m^2^, (5) persons with motor disorders, (6) persons using ambulatory aids, and (7) persons who have been diagnosed with depression.

Additionally, in order to target general middle-aged and elderly people, we carried out a physical strength test (single-foot standing test with eyes open, vertical jump, grip strength, functional reach, 10 m walking time) beforehand, and calculated subjects’ Fitness Age Score (FAS) according to the method proposed by Kimura et al. [30]. The mean FAS value obtained from past data gathered during the elderly physical fitness assessment administered by the two cities (from a total of 1021 participants) was −0.071. According to the paper by Kimura et al. [30], this value is nearly equivalent to the mean value at 74 years of age (and the mean age of subjects in this study was 74.1 years); among subjects over the age of 60 in this study, only those with higher FAS values than this were excluded as individuals exhibiting high physical strength. Meanwhile, for subjects in their 50s, as the mean FAS value for female aged 55 years was 2.01, subjects with higher FAS values than this were also excluded as high-strength individuals.

The state of experimental food intake during the study period was determined based on the content of subjects’ capsule intake diaries at the end of the study period and the remaining quantity of the experimental food. Subjects who failed to complete 90% of the experimental meals were excluded from analysis.

This study was conducted in accordance with the Declaration of Helsinki and other ethical guidelines pertaining to medical research involving human subjects. After oral and written briefings were provided, the informed written consent of all subjects to participation was obtained on self-executed consent forms. This study was conducted with the approval of the institutional review board of Kyoto University of Advanced Science (formerly Kyoto Gakuen University, Kameoka, Kyoto, Japan.) (approval number: 74) and the ethics committee of Ajinomoto Co., Ltd. (approval number: 2017-023). This study was registered beforehand to Clinical Trial Registry (UMIN-CTR; ID UMIN000032451).

#### 2.1.3. Investigational Food and Dose Method

Experimental food: 200 mg of rapeseed oil containing 1.5 mg of capsinoids (the ratio of capsiate, dihydrocapsiate, and nordihydrocapsiate = 7:2:1) extracted from pepper fruit variety CH-19 Sweet (*Capsicum annuum* L.), purified, and encapsulated in a soft vegetable film capsule. Placebo food: 200 mg of rapeseed oil encapsulated in a soft vegetable film capsule. Capsinoids and placebo capsules were provided from Ajinomoto Co Inc (Tokyo, Japan). Subjects ingested either the experimental food or a placebo twice daily (once between waking and breakfast and again between dinner and bedtime). Each dose was comprised of 3 capsules, so CP subjects ingested 9 mg capsinoids a day. The dosing period was 12 weeks. Capsule intake diaries were provided to each subject, and subjects were instructed to record before going to sleep each day whether they had taken the experimental food.

#### 2.1.4. Outcomes and Assessments Tools

Our primary outcome was physical activity level in daily life, and the secondary outcomes were mood profile, body composition and fatness, and temperature sensitivity. We measured the items below.

(1) Physical activity level in daily life: PA intensity (Metabolic equivalents; METs) was measured using a waist-mounted activity meter with 3-axis accelerometer (Actimarker, EW4800, Panasonic Co., Kadoma-city, Osaka, Japan). This activity monitor was previously validated using the doubly labeled water method in Japanese older adults [31,32].

(2) Mood profile: measured using the Japanese version of the Profile of Mood Status 2 (POMS2) short form [33]. The POMS2 determines 7 subscales (anger-hostility, confusion-bewilderment, depression-dejection, fatigue-inertia, tension-anxiety, vigor-activity, and friendship) and total mood disturbance (TMD) score, which quantify subjects’ negative mood status.

(3) Body composition and fatness: height was measured by a calibrated stadiometer. Body weight, lean body mass (LBM), percent body fat (% fat), and visceral fat were measured and estimated with a multi-frequency standing posture eight-electrode bioelectrical impedance analysis (Yamato Scale Co Ltd., DF860K, Akashi, Hyogo, Japan) in which device weight scale is incorporated. Based on the specification sheet of the DF86K, the LBM and % fat estimated by this multi-frequency Bio-electrical Impedance Analysis (BIA) were highly correlated with LBM and % fat obtained from dual X-ray absorptiometry (*r* = 0.974 and 0.947, respectively). As well, visceral fat index obtained by this device was highly correlated with the visceral fat cross-sectional area (CSA) of abdominal CT scanning (*r* = 0.857). The value of visceral fat index (e.g., 100) corresponds to a visceral fat CSA (e.g., 100 cm^2^). The waist circumference was also measured by placing a tape measure horizontally around the abdomen at the umbilical level.

(4) Temperature sensitivity: cold as sensed by the body was measured using a visual analog scale (VAS) consisting of 10 cm line segments. In VAS measure for temperature sensitivity, a 10 cm line was presented to the subjects. On the left end was written “Do not feel cold at all,” and on the right end was written “Feel cold every time.” Subjects were instructed to draw a mark on the point on the line where their body’s cold feeling was adequate between the two extremes. The length between the left end and the mark was measured in 1 mm units.

#### 2.1.5. Measurement Schedule 

The duration of this study was from January to May 2018. Subjects were measured on a total of three occasions. For assessment 1, subjects wore the activity monitor from the day after the initial measurement to the day before the last measurement. The 7 days from the start of activity meter use was designated as the baseline measurement period. Taking of experimental food started from the following day, day 8. The second measurement was carried out at week 6 after the start of dosing and the third and final measurement at week 12. Assessments of 2–4 were measured during the initial measurement and also measured at the second and third measurements. 

#### 2.1.6. Randomization and Blinding

The subjects were allocated to the CP or PL group beforehand by random digits generated in a personal computer. Sex distribution was balanced in each group. The experimental foods were all the same in appearance but distinguished by a letter P or Q on the package, each of which stood for one of the two types of experimental foods. All the subjects could not know the study data and allocation information until the study analyses finished. All the investigators could not know the allocation information until the study analyses finished. 

#### 2.1.7. Statistical Analysis for the Human Study 

The number of steps and the physical activity intensity in METs were collected by the well-validated triaxial accelerometer every minute throughout the wearing period. In cases where PA was 1.1 METs or less for longer than 60 min before midnight, we took it as a nonwearing period and excluded it from the data. Periods where the intensity was less than 1.5 METs and the number of steps taken equaled 0 were taken as sedentary/inactive periods. Sedentary time was obtained as the total duration of the sedentary/inactive periods per day. When the accelerometer output indicated 1.5 METs or above, we considered those periods as PA. Physical activity is typically divided into light (1.5 to 2.9 METs), moderate (3.0 to 5.9 METs), or vigorous (6.0 METs or more) intensity. For older adults, time spent on any amount of light PA is significantly and positively correlated to total physical activity level as well as moderate PA [32]. It may be because the cutoff value of 3.0 METs between light and moderate PA is based on young adults and corresponds to absolute intensity of PA, which does not reflect relative intensity with their maximal aerobic capacity in older adults. In older adults who have low maximal aerobic capacity, the PA of 2.5 METs may not be light PA. We therefore combined the duration of light-to-moderate PA (LMPA) for further analysis rather than treating them as separate variables. In contrast, vigorous PA (VPA) is scarcely observed, particularly in middle-aged and older adults. In fact, the majority of the current participants engaged in no or very little VPA based on the output of the accelerometer as well as previous literature [34]. Therefore, we decided to show the time spent on VPA separately from LMPA for confirmation. We obtained the energy expenditure of PA per day using the following formula:Energy expenditure = METs × Time (number of hours) × Body weight (kg) × 1.05

This energy expenditure includes basal metabolic rate (BMR), and BMR was subtracted to obtain the net energy expenditure in the physical activity. BMR was predicted using the previously validated formula by Ganpule et al. to Japanese people [35].

The data for days subjects wore the activity meter for less than 10 h were also excluded as invalid data. Additionally, subjects who fell under any of the following criteria were excluded from the analysis of activity meter data: (1) subjects with 3 or fewer days of valid data during the 7 day baseline period and (2) subjects for whom the number of valid days was less than 80% of the study period, due to nonwearing, equipment failure, or loss.

For each subject, missing values among the activity monitor data were complemented with the maximum value for sedentary/inactive period, and the minimum value for others during the study period. Next, the average value during the baseline (BL), week 1–4 (4 w), week 5–8 (8 w), and week 9–final (12 w) were calculated, and the changes between BL and 4 w, 8 w, and 12 w values were evaluated through a repeated measures analysis of variance (ANOVA) test applying a linear mixed model. Changes of values at 4 w, 8 w, and 12 w compared with the BL were also tested using a covariance analysis, with the BL data serving as covariates.

In the POMS2 short form, VAS data, waist circumference, and body composition recorded during the initial measurement were taken as the baseline (BL), and data recorded during the second measurement at week 6 (6 w) and during the third measurement at week 12 (12 w) were evaluated through a repeated measures ANOVA test applying a linear mixed model. Changes of values at 6 w and 12 w compared with the BL were also tested using a covariance analysis, with the BL data serving as covariates.

Additionally, as a subgroup analysis, we analyzed subjects exhibiting tendencies to inactivity or overweight. For the former subgroup, the median sedentary time was calculated, and subjects whose sedentary time was this value or more were targeted for analysis. Subjects in the latter subgroup were those with a BMI of 23.0 kg/m^2^ or higher, except for those with a BMI of 25.0 kg/m^2^ or higher as well as visceral fat of 100 cm^2^ or greater, indicating obesity, who were excluded from the subgroup analysis.

SPSS Statistics Ver. 25 was used for statistical analysis. Values are shown as mean ± standard deviation. *p*-values less than 0.05 were considered significant.

### 2.2. Animal Study

#### 2.2.1. Animals and Diets

C57BL/6J male mice were purchased from Charles River Japan (Kanagawa, Japan) at 5 weeks and 53 weeks of age and housed in a controlled-lighting environment (lights on from 20:00 to 08:00 h) at 25 ± 1 °C. They were fed a CRF-1 diet (Charles River Japan, Kanagawa, Japan) for 1 week to stabilize their metabolic condition. The mice were then divided into three groups three ways, by age, body weight, and food intake. They were allowed ad libitum access to water and were fed either a diet based on AIN-93G (normal diet, ND) or an ND supplemented with 0.3% (w/w) dihydrocapsiate (DHCte) (Ajinomoto, Kanagawa, Japan). The precise composition of each diet is shown in Table 1. In this way, we established three groups: (1) 5-week-old mice fed an ND (young mice), (2) 53-week-old mice fed an ND (aged mice), and (3) 53-week-old mice fed an ND supplemented with DHCte (aged mice supplemented with DHCte). The mice were maintained on these diets for 12 weeks. Body weight and food intake were monitored daily, and the spontaneous activities of the mice were simultaneously measured using an activity sensor (NS-AS01; Neuroscience, Tokyo, Japan). The measurements were performed at 4.5 min intervals. All animal protocols were approved by the Animal Committee of Ajinomoto Co., Inc.

#### 2.2.2. Sampling Procedures

At the end of the experiment, the mice were anesthetized with isoflurane, and blood samples were collected via aortic puncture after 3 h of fasting. The blood samples were centrifuged at 6000× *g* for 10 min at 4 °C and stored at −80 °C until use. The organs (prefrontal cortex, hypothalamus, and brain except prefrontal cortex and hypothalamus) were excised. The organs were then immediately frozen in liquid nitrogen and stored at −80 °C.

#### 2.2.3. Gene Expression Analysis

Total RNA was isolated using RiboZol (Amresco, Solon, OH, USA) according to the manufacturer’s protocol. Reverse transcription reactions were performed using an iScript cDNA synthesis kit (Bio-Rad). The primer sequences used in the amplification are shown in Table 2. The qRT-PCR was performed with the SYBR Select Master Mix (Life Technologies, Carlsbad, CA, USA) using a QuantStudio 12 K Flex Real-Time PCR System (Thermo Fisher, Hudson, NH, USA). The relative mRNA expression was determined by the ΔΔ-Ct method using the TATA-binding protein (TBP) or 18S ribosomal RNA (18S) as an internal control.

#### 2.2.4. 8-Hydroxy-2-deoxyguanosine (8-OHdG) Assay

Genomic DNA was isolated from extra brain using a DNA Extractor TIS Kit (Wako, Osaka, Japan). The DNA samples were diluted in TE buffer. Then, hydrolysis of the DNA was performed using the 8-OHdG Assay Preparation Reagent Set (Wako, Osaka, Japan). The hydrolysate was filtered through Amicon Ultra (Merck Millipore, Billerica, MD, U.S.A.) prior to the measurement of 8-OHdG using a Highly Sensitive 8-OHdG Check (Nikken SEIL Corporation, Shizuoka, Japan).

#### 2.2.5. Statistical Analysis for the Animal Study

Statistical analysis was performed using JMP version 10.0 (SAS Institute Inc., Cary, NC, USA). Repeated measures ANOVA was applied to compare the time courses; statistical significance was determined by Dunnett’s test. Values are shown as the mean ± standard deviation unless otherwise indicated. Calculated *p*-values less than 0.05 were considered significant. 

## 3. Results

### 3.1. Human Study

#### 3.1.1. Participant Flow

Figure 1 shows the flow and numbers of the participants. Eighty subjects participated in the study; however, during the intervention period, four participants dropped out by disease or injury not caused by the experimental food, and a participant quitted by personal reason. In addition, 22 participants had insufficient data accumulation in activity meter. 

#### 3.1.2. Amount and Time of Physical Activity

Analysis of PA targeted 26 subjects in CP (4 males and 22 females, age: 73.7 ± 6.5) and 21 subjects in PL (5 males and 16 females, age: 73.0 ± 7.9). There were no significant differences in age or the ratio of male and female between the two groups.

First, sex effect was examined with repeated measures ANOVA. For the amount and the time of LMPA, a main effect of sex was found, but there was no sex × period interaction. For the amount and time of VPA, a significant sex × period interaction (*p* = 0.020 for amount and *p* = 0.019 for time) and a main effect of sex (*p* = 0.018 for amount, and *p* = 0.016 for time) were found. Neither significant interaction nor effect was found for the sedentary time and the energy expended in physical activity. Next, repeated measures ANOVA tests applying a linear mixed model were conducted for all participants to find differences between CP and PL in all items except the amount and the time of VPA. A significantly larger change was found in PL compared to CP of the energy expended in physical activity (*p* = 0.010 at 8 w and *p* < 0.001 at 12 w) (Table 3). In the amount and time of VPA, male and female were separately analyzed, and significant larger changes from BL in PL than CP were found at 12 w of both the amount (CP: 0.4 ± 1.3, PL: 0.8 ± 2.6, *p* = 0.025) and the time (CP: 0.5 ± 1.6, PL: 0.9 ± 2.9, *p* = 0.027).

#### 3.1.3. Changes in Amount of Physical Activity in Participants Who Have Inactive Lifestyles

Subjects’ median sedentary time was 468.7 min; thus, 13 subjects in CP (2 males and 11 females, age: 74.4 ± 7.4) and 11 subjects in PL (4 males and 7 females, age: 71.4 ± 9.3) for whom the mean sedentary period was longer than this value were targeted for the analysis. There was no significant difference in age and the ratio of sex between these two groups. No significant sex × period interaction was observed in any of the items. As a result of a linear mixed-model repeated measures ANOVA test, in the CP, the amount of LMPA was found to be significantly more increased at 12 w (*p* = 0.050), and the time associated with LMPA showed significantly larger change at 8 w and 12 w (*p* = 0.020 and *p* = 0.008 respectively) (Table 4, Figure 2A,B). The linear mixed-model repeated measures ANOVA revealed significantly larger changes in energy expended in physical activity of CP compared to PL at all the measurement periods (*p* = 0.033 at 4 w; *p* = 0.002 at 8 w; *p* < 0.001 at 12 w) (Table 4, Figure 2C).

#### 3.1.4. Changes in POMS2 Short Form, VAS and Body Composition

Analyses of POMS2, VAS, and body composition targeted 36 subjects in CP (9 males and 27 females, aged 74.0 ± 7.0) and PL contained 33 subjects (8 males and 25 females, age: 73.6 ± 8.2). There were no significant differences in age and the ratio of males and females between the two groups. No significant sex effect was found in POMS2 and VAS items. During the POMS2 short form evaluation, a significant decrease in the 6 w data for the CP was observed in the anger-hostility subscale scores, only as a result of a linear mixed-model repeated measures ANOVA test (compared to BL; CP: 1.6 ± 3.1, PL: 0.3 ± 2.1, *p* = 0.006), but no significant differences were observed between groups with respect to any other evaluation item. While there were no significant differences in overall VAS, only subjects in the CP aged over 80 years exhibited a tendency for lower VAS during the 6 w covariance analysis for chilly sensation (low “chilly sensation” analog) (CP: 3.7 ± 2.7, PL: 5.9 ± 1.2, *p* = 0.054, η^2^ = 0.257). 

In the body composition analysis, sex effect was examined with repeated measures ANOVA and significant sex × period interaction was found in visceral fat index (*p* = 0.046). Then, all the evaluation items related to body composition except visceral fat index were examined with linear mixed-model repeated measures ANOVA (Table 5). Subjects in CP exhibited a significant decrease in waist circumference at 6 w (*p* = 0.003). Following a covariance analysis using the BL as a covariate, subjects in the CP exhibited significantly lower body fat percentage values at 12 w (*p* = 0.022) (Figure 3A). Males and females were separately analyzed for visceral fat index, whereas significant differences between CP and PL in neither male nor female were found in both ANOVA and ANCOVA. When all the subjects were analyzed together, CP also exhibited a significant decrease in visceral fat index value at 6 w (*p* = 0.016).

#### 3.1.5. Changes in the Body Composition of Overweight Subjects

There were 13 subjects in the CP (2 males and 11 females, age: 73.3 ± 6.6) and 14 subjects in the PL (4 males and 10 females, age: 71.0 ± 9.6) who were targeted for the overweight tendency subgroup analysis. There were no significant differences in age and the ratio of sex between these two groups. As a result of a linear mixed-model repeated measures ANOVA test (Table 6), the subjects from the CP exhibited significantly lower body fat percentage values at 12 w (*p* = 0.026). A covariance analysis using the BL as a covariate also revealed that subjects in the CP had significantly lower body fat percentage values at 12 w (*p* = 0.031) (Figure 3B).

#### 3.1.6. Safety Evaluation

In this study, no adverse events occurred relating to investigational food. There were no safety problems when ingested.

### 3.2. Animal Study

While locomotive activity in light phase (inactive time for mice) was the same between the three groups, locomotive activity in dark phase (active time for mice) was significantly lower in the aged mice than in the young mice. Furthermore, locomotive activity in the DHCte-supplemented mice was higher than that in the aged mice fed a normal diet (Figure 4A). 

Next, we measured the mRNA expression levels of inflammation-related genes in the brain. The expression of various genes related to inflammation, such as C-C Motif chemokine ligand 3 (Ccl3), intercellular adhesion molecule-1 (Icam1), C-X-C motif chemokine 10 (Cxcl10), and interleukin-6 (Il6), was significantly higher or tended to be higher in the aged mice than in the young mice in the prefrontal cortex. Similarly, various inflammatory genes significantly increased associated with aging in hypothalamus. The DHCte supplementation significantly decreased upregulation of the aging-related genes to the same level as that in the young mice, in both prefrontal cortex and hypothalamus (Figure 4B,C). There are reports associating oxidative stress with aging, and oxidative stress can lead to chronic inflammation [36,37]. Hence, we measured the level of 8-OHdG, an oxidative stress marker, in the brain. In aged mice, the level of 8-OHdG in the brain tended to be higher than in young mice (Figure 4D); however, DHCte supplementation returned 8-OHdG levels to those observed in the young mice. 

Lastly, we measured the gene expression levels of antioxidant enzymes in prefrontal cortex and hypothalamus and found that mRNA expression levels of catalase (Cat) in the prefrontal cortex and superoxide dismutase-2 (Sod2) in the hypothalamus was significantly increased by DHCte supplementation, while their expression was not changed by aging (Figure 4E).

## 4. Discussion

In this study, we investigated our hypothesis that continuous ingestion of capsinoids for 3 months would increase PA in middle-aged and elderly people as measured using an activity meter. We especially focused on LMPA because it covers almost all the daily PA in older adults [32,34]. However, we failed to find a significant effect of capsinoids on LMPA among our participants. Individual differences in the physical activity levels of the subjects were large; for example, the mean value of time attributable to baseline LMPA per day varied from 233 min to 606 min across all participants. It is possible that this variation may have obscured the effects of capsinoids ingestion. Thus, we focused our analysis on participants found to spend long periods in sedentary positions, that is, inactive participants. As a result of this subgroup analysis, we found that subjects from the CP exhibited significantly higher times and amounts of LMPA than subjects from the PL. We calculated and checked the energy expended in physical activity in both groups and confirmed that the increased LMPA was related to increased energy expenditure in CP from inactive participants.

Several reports from past studies using activity monitors on Japanese older adults stated that physical activities of 6 METs or more are observed scarcely in daily life. In our study, VPA with 6.0 METs or more was not observed at all in 33 subjects, and the mean activity period per day at this level was just 0.5 min. It can be inferred based on this observation that LMPA, which is the sum of all light- and moderate-intensity activity, comprises nearly the total amount of daily physical activity of the subjects in this study. Stated differently, the above results indicate that the total amount of physical activity per day increased in CP subjects who exhibited an inactive tendency.

Numerous papers have agreed on the importance of moderate physical activity (3.0 METs or more and less than 6.0 METs) to maintaining physical functions, preventing diseases, and reducing mortality rates [5]. Meanwhile, the value of the contribution of light physical activity to health has become recognized in recent years [32,38]. As a result of a several studies conducted by categorizing low-intensity physical activity of less than 3.0 METs into either low-intensity light physical activity (LLPA) of 1.6 METs or more and less than 2.0 METs, or high-intensity light physical activity (HLPA) of 2.0 METs or more and less than 3.0 METs, HLPA has been reported to improve insulin resistance, activity as mild as walking has been reported to have led to reduced severity of stroke events, and these activities have been partially correlated with physical functions in elderly Japanese women [38,39,40]. Furthermore, a study by Buman et al. [41] demonstrated that HLPA contributes to health to the same degree as moderate physical activity and also contributes greatly to subjective life satisfaction. Even more interestingly, this study also found that replacing just 30 min of sedentary time with light physical activity can lead to improvements in both overall health and subjective lifestyle assessment. Although no reduction in sedentary time was observed in the CP subjects in this study who exhibited inactive tendencies, the significant increase in the time and amount of LMPA in these same subjects suggested that continuous ingestion of capsinoids led to improvements in physical functions, overall health, and thus subjective lifestyle satisfaction.

In our animal research to elucidate the relation between brain inflammation and PA, we found that old mice were less active than young mice, whereas capsinoids supplementation improved age-related reduction of PA in animals as well as humans (Figure 4A). 

Capsinoids are reported as a potent anti-inflammatory compound that targets specific immune pathways and reduces the extensive damage of inflammation in mice [25]. In our animal study, as shown in Figure 4A–C, old mice with ND indicated less activity and aging-induced increase of mRNA levels related to inflammation in the brain, whereas old mice with DHCte were as active as young mice, and inflammation-related mRNA levels were suppressed. The anti-inflammatory effect of DHCte in the brain, not only in peripheral tissues, can at least partly contribute to improving physical activity in older mice, and it may also explain the increased LMPA in inactive older adults. We hypothesized that the alleviation of brain inflammation would appear as positive change of mood state; however, it was unclear. The inflammation in the brain is presumed to be responsible for age-related brain deteriorations; therefore, the possible function of capsinoids as an anti-inflammation ligand needs further investigation.

Another possible interpretation is that the capsinoids were responsible for the change in energy production. Both one-time ingestion and 2 week continuous ingestion of capsinoids suppressed glucose use by increasing oxidative phosphorylation in muscles and facilitating lipid use [42,43]. Additionally, mice that ingested capsinoids exhibited longer endurance swimming capacity, implying that capsinoids-induced change in energy production increases endurance exercise capacity [44]. To relate these findings to the present study, capsinoids ingestion may have supported sustained physical exercise in the elderly subjects, thereby reducing their reluctance to physical activity and their sense of fatigue.

Based on the results of analysis of the body-composition-related evaluation items, significant decreases were observed in CP subjects with respect to waist circumference, visceral fat index, and body fat percentage. Moreover, our subgroup analysis focusing on overweight subjects (excluding obese subjects) also found significant decreases in body fat percentage. These findings are consistent with previous studies and are considered to be the result of the lipid-oxidation-enhancing physiological effect of capsinoids [18,45].

Continuous ingestion of capsinoids has been reported to promote the formation of brown adipose tissue and thereby to accelerate thermogenesis, which may lead to alleviation of complaints of chills [19,46]. Chills are a common complaint in Japanese people: 53.1% of men and 81.6% of women aged 65 or older, as well as 71.0% of men and 95.2% of women aged 75 or older, have complained of cold hands and feet (Japanese Ministry of Health, Labour and Welfare, 2010). Takagi et al. [47] conducted a study focused on thermogenesis and reported that women who chronically felt cold had lower basal energy consumption and sympathetic nervous activity related to body temperature regulation and heat production. Additionally, another study found that young men and women who ingested the CH-19 sweet pepper directly exhibited increased sympathetic nervous activity and elevated tympanic membrane temperature, so capsinoids ingestion may relieve these causes of chills [48]. In the present study, subjects over 80 years of age tended to feel reduced chilly sensation. In another study, following an intravenous injection of physiological saline cooled to 4 °C, middle-aged and elderly subjects exhibited lower concentrations of norepinephrine in the blood, less oxygen intake increase, and less blood flow decrease to the fingertips compared to younger subjects, resulting in lower body core temperatures than the younger subjects [49]. This result suggests that age-related deterioration in the capacity of homeostasis may be responsible for the chilly sensation, and this is consistent with the observation of more frequent complaints of chills in older people. The results of our study suggest that enhancement of sympathetic nervous activity and subsequent thermogenic effects due to capsinoids ingestion were especially noticeable in subjects aged 80 years or older who experienced particularly lowered homeostatic capacity. This study investigated the effects of capsinoids on cold sensitivity in middle-aged and elderly subjects for the first time; however, further study will be necessary in the future.

We have considered that our sample size of the human study was adequate, but there were many deficiencies in the data. It possibly caused the results of all participants to not be very clear. Further investigations with subjects of sufficient number and similar properties would be needed.

## 5. Conclusions

In this study, we investigated changes resulting from continuous ingestion of capsinoids in middle-aged and older adults. The results indicated, with regard to body composition, decreases in waist circumference, visceral fat, and body fat percentage, and a tendency for reduction in chilly sensation in subjects over 80 years of age. In subjects exhibiting tendencies towards inactivity, increased amount and time devoted to LMPA were observed. In an animal study, older mice treated with DHCte showed improved locomotor activity, and in the brain, they showed decreased inflammation-related gene expressions and oxidative stress markers and increased gene expressions of antioxidant enzymes than older control mice. The increased amount of LMPA, which accounted for almost all of the daily physical activity in participants with sedentary/inactive tendencies in CP group, suggests that continuous ingestion of capsinoids can increase the amount of daily physical activity in people exhibiting inactive tendencies.

## Figures and Tables

**Figure 1 nutrients-12-00212-f001:**
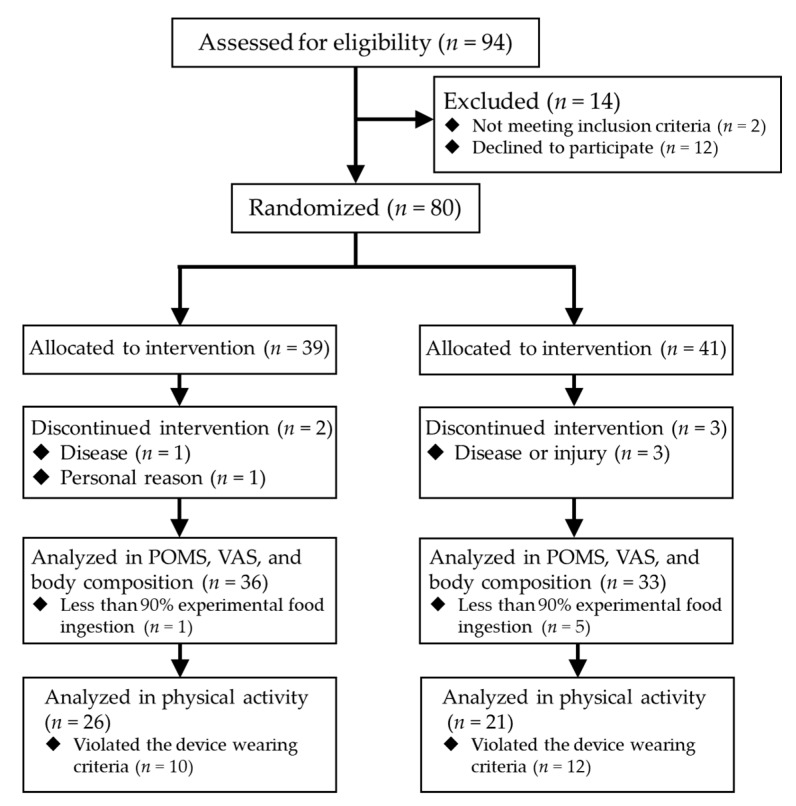
The flow and numbers of the participants.

**Figure 2 nutrients-12-00212-f002:**
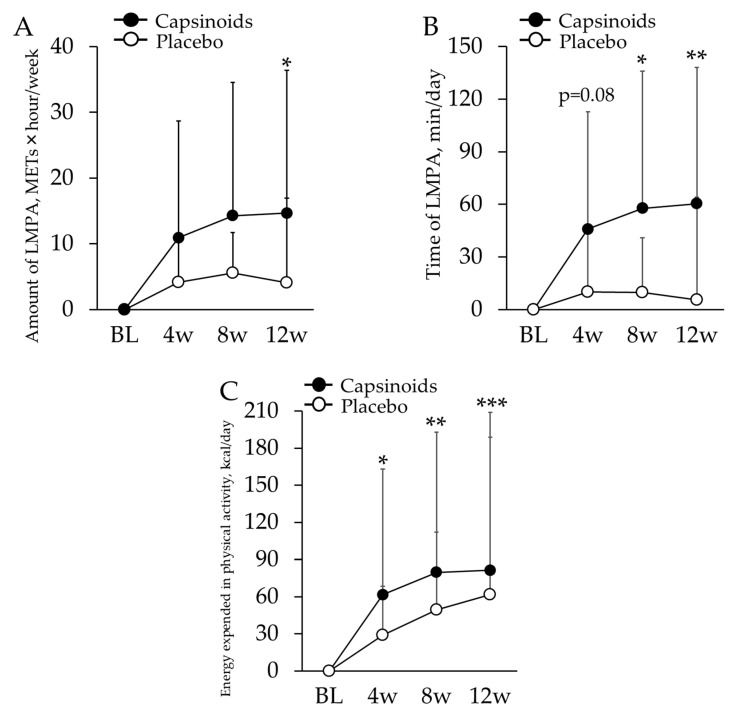
Changes from BL in the amount (**A**) and the time (**B**) of LMPA, and the energy expended in physical activity (**C**) in sedentary/inactive subjects. The values are the difference from baseline value. The error bars indicate standard deviation. Capsinoids group showed significant increase in these measurements of PA (* *p* < 0.05; ** *p* < 0.01; *** *p* < 0.001).

**Figure 3 nutrients-12-00212-f003:**
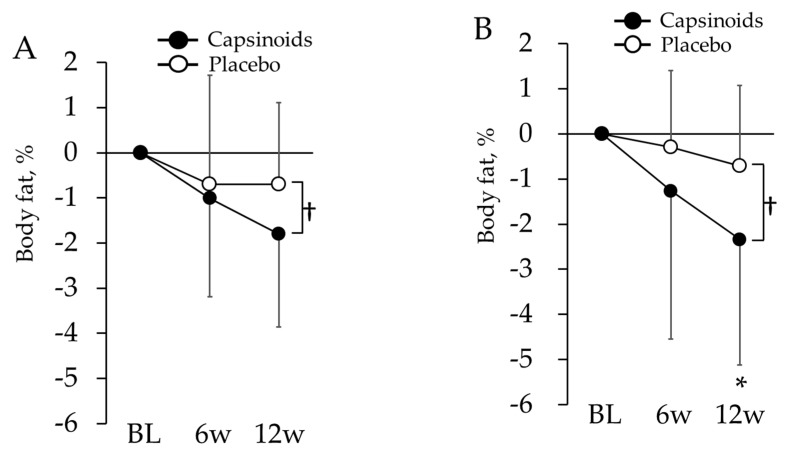
Changes in body fat from BL in all subjects (**A**) and overweight subjects (**B**). The values are the differences from baseline value. The error bars represent standard deviation. Body fat in capsinoids group was lower than placebo group at the 12th week († *p* < 0.05). In overweight subjects, it also showed significant change in capsinoids group compared with placebo group at the 12th week (* *p* < 0.05).

**Figure 4 nutrients-12-00212-f004:**
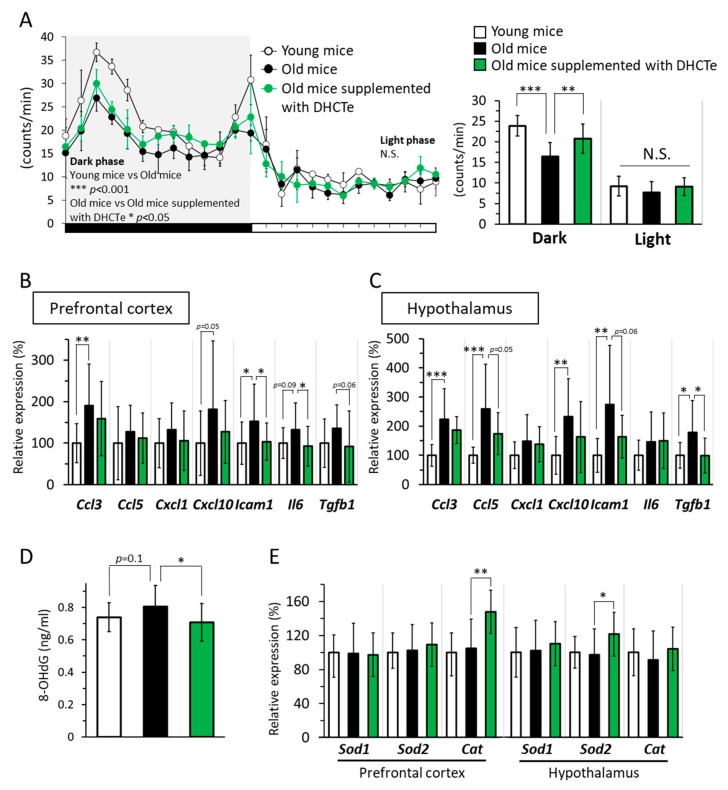
DHCte supplementation suppressed age-associated reduction of physical activity and inflammation and oxidative stress in the brain. (**A**) (Left) The time course of locomotive activity in young and old C57BL/6J mice fed either an ND or an ND supplemented with DHCte for 56 days. (Right) Locomotor activity of mice in young and old mice was measured during 12 h. (**B**) mRNA expression levels of C-C motif chemokine ligand 3 (Ccl3), C-C motif chemokine ligand 5 (Ccl5), C-X-C motif chemokine 1 (Cxcl1), C-X-C motif chemokine 10 (Cxcl10), intercellular adhesion molecule 1 (Icam1), interleukin 6 (Il6), and transforming growth factor-β1 (Tgfb1) in the prefrontal cortex. (**C**) mRNA expression levels of Ccl3, Ccl5, Cxcl1, Cxcl10, Icam1, Il6, and Tgfb1 in the hypothalamus. (**D**) 8-hydroxy-2-deoxyguanosine (8-OHdG) in the brain. (**E**) the mRNA expression levels of superoxide dismutase-1 (Sod1), superoxide dismutase-2 (Sod2), and catalase (Cat) in the prefrontal cortex and hypothalamus of young and old C57BL/6J mice fed either an ND or an ND supplemented with DHCte for 84 days. The values represent the means ± standard deviation (*n* = 14–22). * *p* < 0.05; ** *p* < 0.01; *** *p* < 0.001.

**Table 1 nutrients-12-00212-t001:** Composition of the experimental diets (%).

	ND	ND + DHCte (0.3%)
Casein	20	20
Sucrose	10	10
Cornstarch	39.8	39.8
α-cornstarch	13.2	13.2
l-cysteine	0.3	0.3
Cellulose	5	5
Lard	7	7
Mineral mix	3.5	3.5
Vitamin mix	1	1
Choline bitartrate	0.25	0.25
Dihydrocapsiate	-	0.3

**Table 2 nutrients-12-00212-t002:** Primers used in the real-time PCR analysis.

Gene	Sense	Antisense	Entrez Gene ID
*Cat*	AGCGACCAGATGAAGCAGTG	TCCGCTCTCTGTCAAAGTGTG	12359
*Ccl3*	TTCTCTGTACCATGACACTCTGC	CGTGGAATCTTCCGGCTGTAG	20302
*Ccl5*	GCTGCTTTGCCTACCTCTCC	TCGAGTGACAAACACGACTGC	20304
*Cxcl1*	CTGGGATTCACCTCAAGAACATC	CAGGGTCAAGGCAAGCCTC	14825
*Cxcl10*	CCAAGTGCTGCCGTCATTTTC	GGCTCGCAGGGATGATTTCAA	15945
*Icam1*	GTGATGCTCAGGTATCCATCCA	CACAGTTCTCAAAGCACAGCG	15894
*Il6*	TCCAGCCAGTTGCCTTCTTGG	TCTGACAGTGCATCATCGCTG	16193
*Sod1*	AACCAGTTGTGTTGTCAGGAC	CCACCATGTTTCTTAGAGTGAGG	20655
*Sod2*	CAGACCTGCCTTACGACTATGG	CTCGGTGGCGTTGAGATTGTT	20656
*Tgfβ1*	CTCCCGTGGCTTCTAGTGC	GCCTTAGTTTGGACAGGATCTG	21803
*Tbp*	ACCCTTCACCAATGACTCCTATG	TGACTGCAGCAAATCGCTTGG	21374
*18S*	GTGGAGCGATTTGTCTGGTT	AACGCCACTTGTCCCTCTAA	19791

**Table 3 nutrients-12-00212-t003:** Amount of vigorous and light-to-moderate physical activity and time spent in vigorous, light-to-moderate, and sedentary physical activity in all subjects.

	CP (*n* = 26)				PL (*n* = 21)			
	BL	4 w	8 w	12 w	BL	4 w	8 w	12 w
Amount of VPA, METs × h/week	0.7 ± 2.8	0.5 ± 1.5	0.6 ± 1.7	0.9 ± 2.9	0.0 ± 0.0	0.2 ± 0.3	0.8 ± 3.0	2.7 ± 9.4 ^‡^
Time of VPA, min/day	0.9 ± 3.7	0.5 ± 1.8	0.7 ± 2.1	1.1 ± 3.7	0.0 ± 0.1	0.2 ± 0.4	1.0 ± 3.9	3.3 ± 11.8 ^‡^
Amount of LMPA, METs × h/week	103.0 ± 28.2	108.0 ± 26.2	108.3 ± 28.6	108.2 ± 28.3	104.6 ± 19.8	108.8 ± 20.7	112.2 ± 20.0	115.2 ± 23.6
Time of LMPA, min/day	395.3 ± 105.2	418.5 ± 92.6	419.9 ± 101.1	420.4 ± 103.2	407.0 ± 79.1	419.8 ± 83.1	428.2 ± 82.7	440.6 ± 95.1
Sedentary time, min/day	477.5 ± 100.5	469.4 ± 98.1	486.9 ± 118.3	496.9 ± 122.2	464.1 ± 82.6	478.3 ± 92.1	477.2 ± 101.4	460.0 ± 115.9
Energy expenditure in physical activity, kcal/day	594.6 ± 167.1	619.7 ± 168.8	617.5 ± 154.3	619.9 ± 155.5	565.0 ± 115.6	593.3 ± 126.0	620.3 ± 126.7 ^†^	653.3 ± 166.3 ^†††^

Abbreviations: CP, capsinoids group; PL, placebo group; BL, baseline; 4 w, 4th week; 8 w, 8th week; 12 w, 12th week; VPA, vigorous physical activity; LMPA, light-to-moderate physical activity. Significantly larger change than CP in linear mixed model repeated ANOVA (^†^
*p* < 0.05; ^†††^
*p* < 0.001); significantly larger change than CP in linear mixed model repeated ANOVA in females (^‡^
*p* < 0.05).

**Table 4 nutrients-12-00212-t004:** Amount of vigorous and light-to-moderate physical activity and time spent in vigorous, light-to-moderate, and sedentary physical activity in inactive subjects.

	CP (*n* = 13)				PL (*n* = 11)			
	BL	4w	8w	12w	BL	4w	8w	12w
Amount of VPA, METs×hour/week	0.0 ± 0.0	0.0 ± 0.0	0.1 ± 0.2	0.0 ± 0.0	0.0 ± 0.1	0.2 ± 0.3	1.3 ± 4.1	4.2 ± 12.8
Time of VPA, min/day	0.0 ± 0.1	0.0 ± 0.0	0.1 ± 0.2	0.0 ± 0.0	0.1 ± 0.1	0.3 ± 0.4	1.8 ± 5.4	5.2 ± 16.1
Amount of LMPA, METs×hour/week	84.5 ± 17.2	95.4 ± 21.1	98.8 ± 22.8	99.2 ± 24.9*	99.7 ± 23.3	103.8 ± 23.5	105.2 ± 20.8	103.8 ± 21.9
Time of LMPA, min/day	329.0 ± 63.0	374.8 ± 72.5	386.6 ± 78.1 *	389.4 ± 84.2 **	385.6 ± 94.2	395.6 ± 93.6	395.3 ± 84.8	391.0 ± 86.3
Sedentary time, min/day	558.9 ± 62.9	532.9 ± 81.8	559.2 ± 93.6	576.9 ± 90.7	524.4 ± 57.7	541.3 ± 57.8	544.5 ± 75.2	533.3 ± 89.0
Energy expenditure in physical activity, kcal/day	481.2 ± 96.3	542.7 ± 127.1 *	560.9 ± 129.5 **	562.5 ± 145.5 ***	536.8 ± 112.2	565.7 ± 109.7	586.4 ± 101.7	598.6 ± 127.6

Abbreviations: CP, capsinoids group; PL, placebo group; BL, baseline; 4 w, 4th week; 8 w, 8th week; 12 w, 12th week; VPA, vigorous physical activity. Significantly larger change than PL in linear mixed model repeated ANOVA (* *p* < 0.05; ** *p* < 0.01; *** p < 0.001).

**Table 5 nutrients-12-00212-t005:** Physical attributes in all subjects.

	CP (*n* = 36)			PL (*n* = 33)		
	BL	6 w	12 w	BL	6 w	12 w
Body weight, kg	57.2 ± 8.2	57.2 ± 8.4	57.0 ± 8.1	56.3 ± 7.5	56.3 ± 7.3	56.3 ± 7.3
BMI, kg/m^2^	23.3 ± 2.3	23.3 ± 2.4	23.2 ± 2.5	23.5 ± 2.8	23.5 ± 2.8	23.6 ± 2.7
Abdominal circumference, cm	86.0 ± 7.5	85.4 ± 7.0 **	86.3 ± 7.1	83.3 ± 8.6	85.0 ± 8.1	84.0 ± 8.3
Percent body fat, %	35.0 ± 6.5	34.0 ± 6.3	33.2 ± 6.3 ^†^	34.9 ± 7.2	34.2 ± 8.1	34.2 ± 7.3
Visceral fat index	103.8 ± 31.6	102.2 ± 25.4 *	103.9 ± 24.1	98.9 ± 25.2	102.7 ± 24.4	98.2 ± 24.8
Muscle mass, kg	14.1 ± 2.6	14.2 ± 2.1	14.3 ± 2.1	14.1 ± 1.9	14.3 ± 2.1	14.2 ± 2.1
Muscle mass percentage, %	25.1 ± 2.4	25.2 ± 2.3	25.4 ± 2.4	25.1 ± 2.7	25.5 ± 3.0	25.3 ± 2.5
Fat free mass, kg	36.1 ± 8.6	33.9 ± 12.9	38.0 ± 6.1	36.5 ± 5.6	34.8 ± 10.3	36.9 ± 5.7

Abbreviations: CP, capsinoids group; PL, placebo group; BL, baseline; 6 w, 6th week; 12 w, 12th week. Significantly larger change than PL in linear mixed model repeated ANOVA (* *p* < 0.05; ** *p* < 0.01); significant difference between CP and PL in ANCOVA (covariate = baseline value) (^†^
*p* < 0.05).

**Table 6 nutrients-12-00212-t006:** Physical attributes in overweight subjects.

	CP (*n* = 13)			PL (*n* = 14)		
	BL	6 w	12 w	BL	6 w	12 w
Body weight, kg	59.2 ± 4.9	59.0 ± 5.0	58.8 ± 5.0	58.4 ± 4.5	58.3 ± 4.7	58.3 ± 4.7
BMI, kg/m^2^	23.9 ± 0.6	23.8 ± 0.8	23.8 ± 0.9	24.6 ± 1.4	24.6 ± 1.5	24.7 ± 1.4
Abdominal circumference, cm	87.6 ± 4.6	87.2 ± 5.3	88.5 ± 4.3	85.1 ± 5.1	87.0 ± 4.0	86.2 ± 4.9
Percent body fat, %	37.7 ± 5.4	36.5 ± 4.8	35.4 ± 4.1 *^,†^	37.0 ± 7.0	36.7 ± 6.7	36.3 ± 7.0
Visceral fat index	99.6 ± 28.8	100.0 ± 19.3	102.3 ± 20.6	100.4 ± 25.6	105.0 ± 23.6	100.7 ± 23.5
Muscle mass, kg	37.0 ± 5.8	37.6 ± 5.8	38.1 ± 5.3	36.9 ± 5.8	37.0 ± 5.9	37.3 ± 6.1
Muscle mass percentage, %	14.1 ± 3.4	14.8 ± 1.9	14.7 ± 1.9	14.6 ± 2.3	14.7 ± 2.4	14.7 ± 2.4
Fat free mass, kg	24.7 ± 2.2	25.0 ± 1.8	25.1 ± 2.2	24.9 ± 3.0	25.2 ± 2.8	25.2 ± 2.8

Abbreviations: CP, capsinoids group; PL, placebo group; BL, baseline; 6 w, 6th week; 12 w, 12th week. Significantly larger change than PL in linear mixed model repeated ANOVA (* *p* < 0.05); significant difference between CP and PL in ANCOVA (covariate = baseline value) (^†^
*p* < 0.05).

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
