# Peer review of "Effects of Capsinoids on Daily Physical Activity, Body Composition and Cold Hypersensitivity in Middle-Aged and Older Adults: A Randomized Study"

_nutrients, 2020, doi:10.3390/nu12010212_

Round 1

Reviewer 1 Report

This paper describes the effects of extracts of capsinoids on daily physical activity, body composition and cold hypersensitivity in middle-aged and older adults. The work is very complete, well designed, and provides great information on the effect of capsinoids in humans. The authors have a great knowledge of the subject as it is observed in the bibliography, and this work deepens even more in this field. On the other hand authors should revise some English expressions along the manuscript.

Observations:

Affiliation: Affiliation number 8 do not appears in authors names “lines 5-7”.

Lines 32, 37, 38, 39,….: Put separations after and before “±”, “<”, “=”, . Unify and apply to the entire document.

Lines 32, 147: “ 9 mg of capsinoids”… ¿What capsinoids?. An extract, synthetic capsinoids, purified capsinoids. In what proportion?. Specify which capsinoids are administered.

Lines 35, 41, …..: Use DHCte; It is the commonly accepted way of abbreviating dihydrocapsiate. Unify and apply to the entire document.

Lines 37, 38, 39, 284,…..: Put “p” in italics. Unify and apply to the entire document.

Lines 52, 55, 60….: Put the references “[x]” before the “.” according the journal format. Unify and apply to the entire document.

Lines 101, 138, 150…: Capitalize each word according journal format. Unify and apply to the entire document.

Lines 101, 138, 150…: Do not put in italics (second order section). Use journal format. Unify and apply to the entire document.

Line 119: Define BMI (body mass index).

Line 139: : “ 1.5 mg of capsinoids”… ¿What capsinoids?. An extract, synthetic capsinoids, purified capsinoids. In what proportion?. Specify which capsinoids are administered.

Line 158: Define TMD.

Line 173: Define CSA.

Line 246: Define DTC.

Line 254, 262, …. All tables: Revise table format according the format of the journal. Unify and apply to the entire document.

Tables 3, 4, 5, 6: (n = 26…)… Put “n” in italics. Unify and apply to the entire document.

Lines 307, 317, …..: Write “Figure” instead of “Fig” according journal format. Unify and apply to the entire document.

References: Use the correct format for journal names according the journal format.

Reviewer 2 Report

While the article is potentially interesting the authors do not adhere to instructed advocacy of reporting by CONSORT Statement standards. (see Instructions to Authors)

These omissions and others factors are noted below and the authors are encouraged to read the checklist they were asked to submit. Unfortunately, these would have to be changed ex post facto or after the fact, which is not how the scientific model works.

[Comment]  There are no stated primary and secondary outcomes. Since this is not stated one can only assume that the investigators did not consider the relative importance of their outcomes in their experimental model and would have to identify this after the fact. Unfortunately, the lack of priority for multiple outcomes lends it self to a fishing expedition with a high likelihood of spurious findings of significance. I am sure this was not the intention of the authors.

[Comment]  The abstract should report "real data" instead of just p-values for pre-post measures based on pre-specified primary and secondary outcomes.

[Comment]  There is no stated and directed hypothesis.

[Comment]  The authors use SEM in their figures. These should be changed to SD. In general, data should be reported as mean (SD) for pre/post data and 95% CI for mean/percent change. SEM should be avoided all together.

ationale and Advocacy. SEM is a measure of inference and SD is a measure of variance. Statistical Inference is an estimate of the characteristics or properties of a population, derived from the analysis of a sample drawn from it. Standard deviation shows how much individuals within the same sample differ from the sample mean and how the variance surrounding a treatment is best accounted for.

http://www.ncbi.nlm.nih.gov/pmc/articles/PMC1255808/

http://www.ncbi.nlm.nih.gov/pubmed/12644429

http://www.ncbi.nlm.nih.gov/pmc/articles/PMC3487226/

http://www.ncbi.nlm.nih.gov/pmc/articles/PMC2959222/

http://bja.oxfordjournals.org/content/90/4/514.long

See CONSORT Statement. Page 8, 16, 18

[Comment]  You have not presented any effect sizes. EFFECT SIZES should reported for all papers as follows: t-tests = Cohen’s D; ANOVA or GLM  = Partial eta squared.. 

Please see: http://imaging.mrc-cbu.cam.ac.uk/statswiki/FAQ/effectSize?fbclid=IwAR3pZl_6nCEQ9HZEvJrub8HP6_-z4waoHU6tW8P2BKxpV_He3sVsBj7f6PU

[Comment]  You have made no attempt to statistically examine whether or not there was a gender effect. This cannot be assumed based on the ratio of men to women in each treatment arm.

Reviewer 3 Report

Even if the matter convers aspect of interest, I suggest to reorganize the paper. The authors should focuse the paper on human study and not on animal study.In this order the methodological part and presentation and discussion of human study should be restructured and implemented. 

Author Response

Please see the attatchment.

Round 2

Reviewer 2 Report

The authors have done a good job on revising their paper.

If one looks at the Abstract guidelines they state that reports should present actual data pertaining to the primary outcome in the abstract using real/actual data. Not just verbiage. Not just p-values. Actual data. 

Then as much data as possible following, especially as it pertains to secondary findings. Minimally you should report your PA levels, though this is challenging. You will have to decide this for your selves, though the inactive participants is appealing. Perhaps Baseline and 12 wks only, that way you could present both "all" and inactive subjects.

Author Response

Please see the attatchment.

Reviewer 3 Report

The authors have improved the manuscript that it is now suitable for publication 

Author Response

Please see the attatchment.
